# Current Application of iPS Cells in the Dental Tissue Regeneration

**DOI:** 10.3390/biomedicines10123269

**Published:** 2022-12-16

**Authors:** Sayuri Hamano, Risa Sugiura, Daiki Yamashita, Atsushi Tomokiyo, Daigaku Hasegawa, Hidefumi Maeda

**Affiliations:** 1Department of Endodontology and Operative Dentistry, Faculty of Dental Science, Kyushu University, Fukuoka 8128582, Japan; 2OBT Research Center, Faculty of Dental Science, Kyushu University, Fukuoka 8128582, Japan; 3Department of Endodontology, Kyushu University Hospital, Fukuoka 8128582, Japan; 4DDR Research Center, Faculty of Dental Science, Kyushu University, Fukuoka 8128582, Japan

**Keywords:** iPS cells, dental tissue regeneration, PDLSCs

## Abstract

When teeth and periodontal tissues are severely damaged by severe caries, trauma, and periodontal disease, such cases may be subject to tooth extraction. As tooth loss leads to the deterioration of quality of life, the development of regenerative medicine for tooth and periodontal tissue is desired. Induced pluripotent stem cells (iPS cells) are promising cell resources for dental tissue regeneration because they offer high self-renewal and pluripotency, along with fewer ethical issues than embryonic stem cells. As iPS cells retain the epigenetic memory of donor cells, they have been established from various dental tissues for dental tissue regeneration. This review describes the regeneration of dental tissue using iPS cells. It is important to mimic the process of tooth development in dental tissue regeneration using iPS cells. Although iPS cells had safety issues in clinical applications, they have been overcome in recent years. Dental tissue regeneration using iPS cells has not yet been established, but it is expected in the future.

## 1. Introduction

It is difficult to preserve teeth when teeth and periodontal tissues are severely damaged by caries, trauma, and periodontal disease. Recently, it has been reported that there is a correlation between the number of remaining teeth and oral frailty [1]. Hiltunen et al. demonstrated a significant increase in chewing difficulties, dysphagia, and malnutrition with increasing severity of oral frailty [2]. Moreover, oral frailty has been reported to be associated not only with decreased oral function but also with increased mortality [1]. In other words, tooth loss adversely affects human health and leads to decreased quality of life. Therefore, the development of regenerative medicine for tooth and periodontal tissue is desired.

Induced pluripotent stem cells (iPS cells) were generated from mice in 2006 and from humans in 2007 [3,4,5]. It has been reported that iPS cells can be established by introducing four reprogramming factors into somatic cells, such as skin and hair, and have self-renewal potential and pluripotency (Figure 1). Yamanaka et al. identified four factors—OCT3/4, Sox2, Klf4, and c-Myc—necessary for the reprogramming of somatic cells among the 24 factors required for the maintenance of embryonic stem cells (ES cells), and succeeded in generating iPS cells for the first time [4]. Unlike ES cells, there are no ethical issues involved in the use of iPS cells. In addition, unlike somatic stem cells such as mesenchymal stem cells (MSCs), iPS cells have no limit to the cells they can differentiate; thus, iPS cells were expected to be applied clinically. In this article, we will review the current application of iPS cells in dental tissue regeneration.

## 2. Clinical Application of iPS Cells for Disease

iPS cells are applied clinically in two ways: direct cell transplantation and indirect drug discovery using iPS cells. Regarding the transplantation of iPS cells, in 2014 the first in-human trial of iPS cell autologous transplantation was performed on a patient with age-related macular degeneration. As undifferentiated iPS cells form teratomas, iPS cells were induced to differentiate into retinal pigment epithelium cells and transplanted into the patient. At one year after operation, the transplanted cell sheet was intact and the patient’s vision was improved and maintained. Furthermore, no tumor formation or graft versus host disease was observed [6]. In addition, clinical trials using iPS cells have been performed for various diseases, such as heart disease, thrombocytopenia, Parkinson’s disease, spinal cord injury, knee joint cartilage injury, and ovarian cancer [7,8].

Drug discovery applying iPS cells is a method to screen drugs using iPS cells in vitro and to develop the drugs efficiently. iPS cells derived from sick patients reportedly have the ability to reproduce the disease state [9,10,11]. Therefore, elucidation of pathology and development of new drugs have occurred using patient-derived iPS cells. Drugs discovered using iPS cells have been clinically applied to fibrodysplasia ossificans progressiva, spinal muscular atrophy, amyotrophic lateral sclerosis, and Pendred syndrome in Japan. Although iPS cells have been clinically applied in various fields, they have not yet been applied in dental tissues. Currently, many studies of the regeneration of dental tissue are performed using iPS cells.

## 3. Generation of iPS Cells from Dental Tissues

iPS cells are obtained by reprogramming somatic cells, even though human dermal fibroblasts (HDFs) are the most common source for iPS cell generation due to their availability [12]. Other cell types, such as blood cells, neural stem cells, and urine cells have also been successfully reprogrammed into iPS cells [12,13,14].

Kim et al. reported that iPS cells retain the epigenetic memory of reprogrammed somatic cells. The authors generated blood-derived iPS cells and fibroblast-derived iPS cells and assessed their hematopoietic colony formation ability. Blood-derived iPS cells formed more hematopoietic colonies than fibroblast-derived iPS cells [15]. In another report, human pancreatic beta cell-derived iPS cells demonstrated an increased ability to differentiate into insulin-producing cells both in vitro and in vivo compared with isogenic pancreatic non-beta cell-derived iPS cells [16]. These results suggest that the epigenetic memory of donor cells might be advantageous for differentiation into donor cell lineage tissues.

Numerous groups have generated iPS cells derived from dental tissue. iPS cells have been generated successfully from dental pulp cells [17,18], dental pulp stem cells (DPSCs) [19,20], stem cells from human exfoliated deciduous teeth [20], stem cells from apical papilla [19,20], oral mucosa fibroblasts [21], gingival fibroblasts [22,23,24], and periodontal ligament (PDL) cells [24]. Yan et al. demonstrated that the reprogramming efficiency appeared to be greater on dental-derived tissues than on HDFs [20]. In addition, Tomokiyo et al. showed that neural crest cells (NCs) differentiated from human PDL-derived iPS cells had higher multipotency than NCs differentiated from human foreskin fibroblast-derived iPS cells [25]. Although dental tissue might be able to efficiently generate iPS cells and iPS-derived NCs, its ability to regenerate dental tissues has not been clarified; therefore, further studies are needed.

## 4. Application of iPS Cells for Tooth Regeneration

Tooth regeneration research has been conducted based on tooth development. The tooth germ grows as the epithelium begins to invade the neural crest-derived mesenchyme [26]. Nakamura et al. indicated tooth germ epithelial cells play important roles in cusp and root formation, and normal dentin formation [27]. Based on these results, tooth regeneration research focusing on the interaction between dental epithelium and mesenchyme has been promoted.

The cap stage is an important stage in tooth germ development because the enamel organ, dental papilla, and dental follicle that is the origin of periodontal tissue are formed [28,29]. Ikeda et al. demonstrated that a tooth germ was generated by combining E14.5 cap stage epithelium cells and pulp-derived mesenchymal stem cells. In addition, when this tooth germ was transplanted into the alveolar bone in mice, it grew as a normal tooth [30]. However, it is difficult to obtain and maintain enamel epithelial cells, and tooth regeneration research using these cells has been limited [31]. Therefore, Kim et al. established a method for differentiating iPS cells into Hertwig’s epithelial root sheath (iHERS), which plays an important role in the formation of root dentin and cementum. Co-culturing iHERS with DPSCs increased enamel- and dentin-related markers [32]. In addition, Cai et al. differentiated human urine cell-derived iPS cells into epithelial sheets and recombined then with mouse dental mesenchyme cells. These combinations were transplanted into the kidneys of nude mice and formed tooth-like structures with a success rate of up to 30% [13]. However, in the aforementioned studies, the formation of periodontal tissues, such as cementum, alveolar bone, and PDL, was not recognized. In addition, it is difficult to generate large teeth that can be applied to humans. Therefore, regenerative research using iPS cells has also been carried out for each part of the dental tissue, including dental pulp, dentin, enamel, cementum, alveolar bone, and periodontal ligament tissues.

## 5. Application of iPS Cells for Dentin–Pulp Regeneration

Pulpectomy is the first choice when irreversible dental pulp inflammation is caused by caries or trauma. However, there are reports that pulpectomy increases the risk of tooth fracture, leading to future tooth loss [33,34]. Therefore, regeneration of dental pulp tissue is considered to lead to preservation of teeth. Otsu et al. demonstrated that mouse iPS-derived NCs (iPS-NCs) cultured in conditioned medium from mouse dental epithelium cells differentiated into odontoblasts and dental pulp cells [35]. A previous study showed that transfection of paired box 9 (PAX9) and BMP4 genes in iPS-NCs increased the gene expression of dentin matrix protein 1 (Dmp1) and dentin sialophosphoprotein (Dspp) [36]. However, the ability of iPS cells to regenerate dentin–pulp complex not only in vitro but also in vivo was evaluated. Xie et al. assessed whether iPS cells form dentin–pulp complexes in vivo. iPS cells were subcutaneously transplanted into immunodeficient mice with tooth scaffolds. After 4 weeks, tubular dentin was present in the scaffold and DMP1-positive odontoblast-like cells derived from human iPS cells were found around the newly formed dentin structure [37]. In addition, Zhang et al. succeeded in forming functional dentin–pulp complexes derived from iPS-NCs. They subcutaneously transplanted mouse iPS-NCs with Matrigel and tooth scaffolds into the dorsa of mice. DSPP-positive odontoblast-like cells derived from iPS-NCs formed new dentin that was continual with the original dentin of the scaffold. Furthermore, because iPS-NCs also differentiated into endothelial cells, it was suggested that regeneration of dentin–pulp complexes with blood vessels derived from iPS cells was possible. However, one of the iPS-NC/scaffold complexes formed teratomas when they were transplanted in vivo [38]. Tumor formation is caused by contamination with undifferentiated iPS cells, and improvement of the induction protocol before transplantation is required.

Previous studies have revealed that DPSCs play an important role in regeneration of the dentin–pulp complex [39,40,41,42]. DPSCs are a useful cell source for dentin–pulp regeneration; however, there have been no reports of DPSCs generated from iPS cells. Therefore, in the future, it is necessary to induce iPS cell-derived DPSCs and investigate their ability to regenerate the dentin–pulp complex.

## 6. Application of iPS Cells for Amelogenesis

Enamel is the hardest tissue in the human body and plays a role in protecting dentin and dental pulp on the surface of the tooth crown. However, ameloblasts, which are important in the formation of enamel, are lost during tooth eruption and cannot be regenerated once damaged [43]. Therefore, the ability of iPS cells to differentiate into ameloblasts has been investigated. Arakaki et al. reported for the first time the potential of iPS cells to differentiate into ameloblasts. iPS cells cultured on a dental epithelial cell line exhibited increased expression of ameloblastin, an enamel matrix protein [44]. Li et al. assessed the capacity of iPS cells to differentiate into ameloblast-like cells using ameloblast serum-free conditioned medium (ASF-CM). iPS cells were cultured in ASF-CM and subcutaneously transplanted into the dorsa of mice. ASF-CM-treated iPS cells expressed ameloblastin and amelogenin, another enamel matrix protein, in vivo. In addition, when iPS cells were cultured in ASF-CM supplemented with BMP4, they differentiated into not only ameloblast-like cells but also odontoblast-like cells [45]. These results demonstrate that ameloblasts could be differentiated from iPS cells. In the future, it will be necessary to examine whether it is possible to regenerate enamel using iPS-derived ameloblasts.

## 7. Application of iPS Cells for Cementum Regeneration

The tooth is able to function properly with the support of the periodontal tissue. Periodontal tissue is composed of four parts: PDL tissue, alveolar bone, cementum, and gingiva. As cementum, alveolar bone, and PDL tissue play important roles in the implantation of teeth, their regeneration is desired. First, we will describe the regeneration of cementum using iPS cells. Several studies reported that when iPS cells were transplanted into a periodontal defect model, cementum was regenerated together with PDL tissue and alveolar bone [46,47]. However, there are only two reports that focus on the differentiation of cementum from iPS cells. Yin et al. generated embryoid bodies (EBs) from gingival-derived iPS cells and stimulated them with enamel matrix proteins or growth/differentiation factor-5 (GDF-5). The stimulated groups exhibited increased expression of cementum markers, cementum attachment protein, and osteocalcin compared with the control group [48]. In another study, EBs formed from gingiva-derived iPS cells differentiated into periodontal progenitor cells via stimulation with GDF-5. After iPS-derived periodontal progenitor cells were implanted with hydrogel subcutaneously into the dorsa of mice for 6 weeks, the expression of cementum markers, including cementum protein 1 (CEMP1) and bone sialoprotein, was increased in periodontal progenitor cells derived from iPS cells, compared with the groups that received only hydrogel scaffold [49]. As there are few studies on the regeneration of cementum from iPS cells, further investigation is necessary in the future. In particular, it is necessary to investigate whether cementum that has Sharpey’s fibers is necessary for connecting with the PDL tissue.

## 8. Application of iPS Cells for Bone Regeneration

Several studies that evaluated the bone regeneration ability of iPS cells have been reported [46,50]. Levi et al. revealed that transplantation of human iPS cells with BMP2 regenerated the cranium, even with a small number of cells [50]. Furthermore, Duan et al. transplanted iPS cells into a periodontal fenestration defect model for the first time. They transplanted iPS cells with silk scaffold + enamel matrix derivatives (EMDs) into the periodontal fenestration defect model. In the silk scaffold + EMD + iPS cell transplanted group, the periodontal defect sites exhibited marked new bone formation compared with the groups that received only silk scaffold and silk scaffold + EMD [46]. However, because undifferentiated iPS cells carry the risk of tumorigenesis, clinical application of these cells is impossible in an undifferentiated state. In a recent study, iPS cells have been differentiated into the following three cell types and used in bone regeneration: (1) osteoblast lineage cells directly differentiated from iPS cells, (2) MSCs differentiated from iPS-NCs (iPS-NC-MSCs), and (3) MSCs differentiated from iPS cells (iPS-MSCs). Bone regeneration using each cell type is described below.

First, we will describe bone regeneration using iPS-derived osteoblast lineage cells. Several studies have assessed methods of inducing osteoblastic differentiation of iPS cells using multiple small molecules or growth factors in vitro [51,52]. The ability of iPS-derived osteoblast lineage cells to regenerate bone not only in vitro but also in vivo was evaluated. Chien et al. differentiated iPS cells into osteocyte-like cells using BMP6. They transplanted iPS-derived osteocyte-like cells into a vertical alveolar bone resorption model and found significant bone regeneration in the group transplanted with iPS-derived osteocyte-like cells compared with the group that did not receive cells [47]. Bilousova et al. confirmed that EBs formed from iPS cells differentiated into osteoblast-like cells by culturing them in osteoblast medium in vitro. However, the transplantation of these cells subcutaneously into the dorsa of mice did not form bone-like hard tissue in vivo [53]. In addition, Phillips et al. evaluated four different methods of inducing osteoprogenitor cells from skin fibroblast- and bone marrow stem cell-derived iPS cell lines. They stimulated each iPS cell line with the following four factors: (1) combination of dexamethasone + ascorbic acid, (2) retinoic acid, (3) rapamycin, and (4) bFGF + BMP4. iPS cells stimulated with dexamethasone + ascorbic acid or bFGF + BMP4 formed bone in vivo, regardless of their origin. However, the bone formation frequency of these cells was about 10% [54]. As direct differentiation of iPS cells into osteoblast lineage cells does not result in stable bone regenerative capacity in vivo, other approaches are required for bone regeneration using iPS cells.

As a second approach in bone regeneration using iPS cells, iPS-NC-derived MSCs have been utilized. iPS-NC-MSCs have been obtained by culturing iPS-NCs in the α-MEM/FBS medium [55,56,57]. Previous studies reported that iPS-NC-MSCs were differentiated into osteoblast-like cells in vitro [55,56]. Kikuchi et al. recently developed iPS-NC-MSCs that had the capability to regenerate craniofacial bone in vivo. In addition, this study demonstrated that the iPS-NC-MSCs did not form tumors [57]. These results indicated that MSCs differentiated from iPS-NCs are useful cells for bone regeneration. However, this approach requires a differentiation process to convert iPS cells to iPS-NCs, which is costly and time-consuming. Therefore, it is necessary to investigate the bone regenerative ability of iPS cells directly differentiated into MSCs.

Finally, the ability of iPS-MSCs to aid bone regeneration has also been assessed [58,59,60,61]. iPS-MSCs were obtained by culturing cells released from iPS-derived EBs in MSC differentiation medium [58,59,61]. Wang et al. transplanted iPS-MSCs + calcium phosphate cement into rat cranial defects and iPS-MSCs regenerated new bone. In addition, the authors compared the amount of new bone derived from bone marrow MSCs and umbilical cord MSCs with iPS-MSCs, and found that the bone regeneration capacity of iPS-MSCs was similar to that of both MSC types [59]. Another report showed that when iPS-MSCs + BMP2 antibody complex was subcutaneously implanted into the dorsa of mice, endogenous BMP2 was recruited and bone-like tissue was formed by iPS-MSCs [62]. Hynes et al. reported the regeneration of alveolar bone using iPS-MSCs. They transplanted iPS-MSCs into a rat alveolar bone defect model and clarified that iPS-MSCs have the ability to regenerate alveolar bone in vivo [63]. Overall, these studies demonstrate that iPS-MSCs may be able to regenerate alveolar bone, which may lead to the development of new periodontal tissue regeneration therapies.

## 9. Application of iPS Cells for PDL Tissue Regeneration

Finally, we describe the regeneration of PDL tissue that connects cementum to bone. A previous study assessed the ability of iPS cells to aid periodontal tissue regeneration. Duan et al. transplanted iPS cells with silk scaffold + EMD in a periodontal defect mouse model. The iPS cell transplantation group regenerated the PDL tissues with Sharpey’s fiber-like structures compared with the group that did not receive iPS cells [46]. Recently, iPS cells have been differentiated into somatic stem cells before they were transplanted to decrease the risk of tumorigenesis. As MSCs are known to have a high ability to regenerate PDL tissue [64,65,66], several studies used iPS-derived MSCs for the regeneration of PDL tissue. Hynes et al. reported the significant increase in PDL tissues and alveolar bone in a rat periodontal defect model when iPS-MSCs were transplanted with fibrinogen and thrombin clots, compared with the control group [63]. In addition, Chien et al. investigated the PDL tissue regenerative ability of iPS cells differentiated into osteocytes. They transplanted osteocyte-like cells derived from iPS cells into a rat periodontal injury model with BMP6 and hydrogel scaffolds. As a result, PDL tissue and cementum as well as the three-walled bone defects were regenerated in the group transplanted with iPS-derived osteocyte-like cells + BMP6 compared with the group implanted with scaffold only or scaffold + BMP6 [47]. These results demonstrate that iPS cells are an effective stem cell source for regeneration of periodontal tissue destroyed by periodontal disease.

## 10. Generation of PDLSCs from Human iPS Cells

Many studies have reported that transplantation of PDL stem cells (PDLSCs) regenerated PDL tissue [67,68,69,70]. Therefore, we established a method to differentiate iPS cells into PDLSCs for the first time. As PDLSCs are derived from NCs [71], we first differentiated iPS cells into NCs. We succeeded in differentiating iPS-NCs into PDLSC-like cells (iPDLSCs) by culturing them on the extracellular matrix (ECM) derived from primary human PDL cells (HPDLCs) [72]. iPDLSCs had a fibroblast-like morphology and multipotency (Figure 2A,B). Furthermore, they expressed high levels of MSC markers, and also expressed PDL-related markers (Figure 2C,D). Interestingly, iPS-NCs cultured on ECM of HDF showed lower expression of MSC- and PDL-related markers and pluripotency compared with iPDLSCs. It was speculated that the ECM from HPDLCs contains key proteins that have the ability to induce PDLSCs from iPS-NCs. ECM is known to play a critical role in various stem cell differentiation processes [73,74,75,76]. ECM derived from PDL tissue contains several proteins, including COL, POSTN, fibronectin, and laminin [77,78]. Therefore, we first planned to identify the transcription factor that regulates the production of HPDLC-derived ECM with the ability to induce PDLSCs.

We performed a comprehensive analysis of HPDLCs and HDF genes. We identified PAX9, a transcription factor involved in the development of the craniofacial region, limbs, and teeth that is highly expressed in HPDLCs. iPS-NCs cultured on PAX9-knockdown HPDLC-derived ECM decreased the expression of PDL-related markers and the ability to differentiate into adipocytes. However, there was no effect on the expression of MSC markers, proliferation ability, and osteoblastic differentiation ability in these cells. These results suggested that PAX9 partially regulates ECM production in HPDLCs [79]. Further work will be required to identify other transcription factors that control the production of HPDLC-derived ECM with the potential to differentiate into PDLSCs. Furthermore, it is necessary to identify proteins that have the ability to induce PDLSCs from iPS-NCs among the HPDLC-derived ECM produced by these transcription factors.

Wang et al. also differentiated PDLSCs from iPS-NCs using an alternative method. They succeeded in differentiating iPS-NC-derived PDLSCs under xeno-free conditions using bFGF and ROCK inhibitor [80]. However, the ability of iPS cell-derived PDLSCs to regenerate PDL tissue has not yet been demonstrated in vivo. In the future, it will be necessary to examine this using a periodontal tissue injury model.

## 11. Problems of iPS Cells in Clinical Application

Although iPS cells have already been clinically applied in various fields, there were two problems in clinical application. The first is that there is a risk of tumorigenesis because virus vectors and the oncogene c-Myc are used to generate iPS cells. Therefore, Yu et al. developed a method to establish iPS cells without the use of viruses [81]. Maekawa et al. introduced the transcription factor glis family zinc finger 1, instead of c-Myc, into HDF and succeeded in efficiently producing iPS cells [82]. These methods have made it possible to produce iPS cells with a low tumorigenic risk. In addition, because iPS cells are undifferentiated cells with unlimited proliferative potential, it has been reported that transplantation of iPS cells results in the formation of teratomas [3]. Therefore, in recent years, iPS cells have been differentiated into somatic cells before transplantation to reduce the risk of tumorigenesis. As described above, in dental tissue regeneration, iPS cells were often differentiated into somatic stem cells such as NCs and MSCs before transplantation.

Second, there is a risk of contamination with heterologous cells because mouse embryonic fibroblasts (MEFs) and FBS are used to culture iPS cells. MEFs were essential as feeder cells for the initial subculture of iPS cells. Recently, feeder-free iPS cell culture methods have been established to reduce the risk of contamination with MEFs [83,84]. Gelatin, collagen, fibronectin, vitronectin, and laminin have been used in iPS cell culture under feeder-free conditions. In long-term subculture on gelatin and collagen, iPS cells failed to maintain their pluripotency [85]. However, iPS cells cultured on fibronectin, vitronectin, or laminin were able to maintain an undifferentiated state even after long-term subculturing [86,87,88]. Moreover, Yamasaki et al. cultured iPS cells in a serum-free medium, which demonstrates that these cells can be cultured in a medium without FBS [88]. In summary, we also need to use such highly safe iPS cells in dental tissue regeneration research.

## 12. Summary

Newly produced iPS cells had some problems in clinical application. However, these problems have been overcome by improving iPS cell production and culture conditions. The development of iPS cells that can be used for allogeneic transplantation is also underway [89], and is expected to further expand the clinical applications.

Although the application of iPS cells in dental tissue regeneration research has been discussed in many publications (Figure 3), these cells have not yet been applied clinically. Dental tissues are generated by the combination of epithelium and mesenchyme and are considered difficult to regenerate. However, because a mini-sized tooth-like structure was created using iPS cells, we believe that tooth regeneration can be expected in the future.

We also plan to examine the PDL tissue regeneration ability of iPS cell-derived PDLSCs in vivo, and hope this line of research will lead to the development of new periodontal tissue regeneration therapy using iPS cells.

## Figures and Tables

**Figure 1 biomedicines-10-03269-f001:**
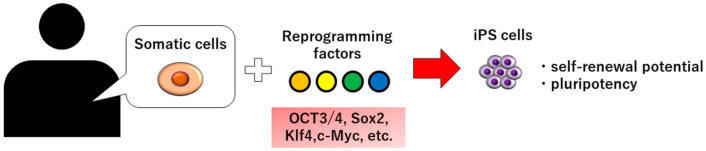
Schematic illustration of iPS cell production.

**Figure 2 biomedicines-10-03269-f002:**
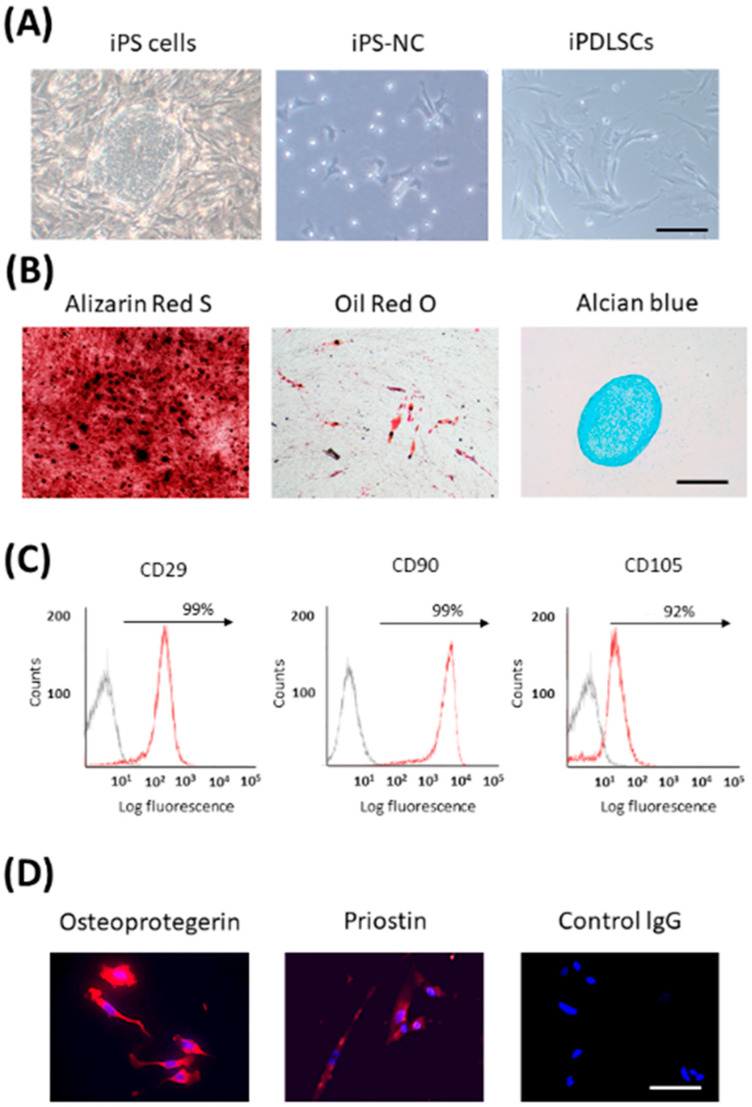
Characterization of iPDLSCs. (**A**) The pictures of iPS cells, iPS-NCs, and iPDLSCs. (**B**) iPDLSCs were cultured in osteogenic, adipogenic, and chondrogenic medium. Mineralization was investigated by Alizarin Red S staining. Fatty lipids were investigated by Oil Red O staining. Cartilage matrix was investigated with Alcian Blue staining. Scale bar = 200 μm. Arrows indicate positive cells. (**C**) The expression of MSC markers (CD29, CD90, and CD105) in iPDLSCs was examined by flow cytometric analysis. (**D**) The expression of PDL-related markers, osteoprotegerin and periostin (red), in iPDLSCs was examined by immunofluorescence staining. Nuclei were stained with DAPI (blue). Scale bars = 200 μm.

**Figure 3 biomedicines-10-03269-f003:**
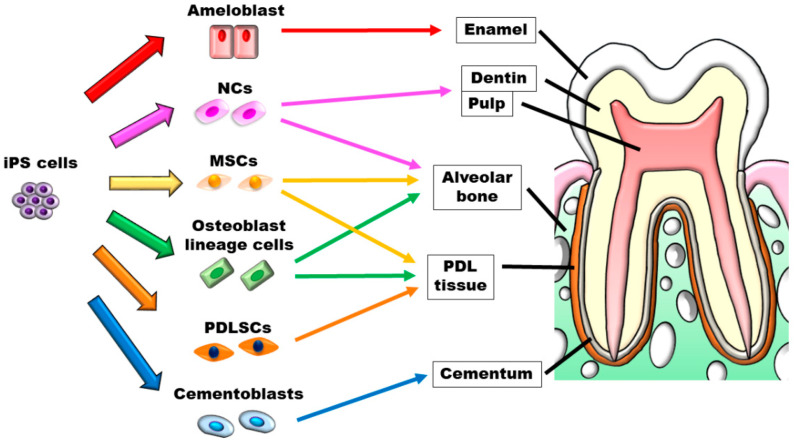
Schematic illustration of iPS cells’ application in dental tissue regeneration.

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
