# Peer review of "Current Application of iPS Cells in the Dental Tissue Regeneration"

_biomedicines, 2022, doi:10.3390/biomedicines10123269_

Round 1
Reviewer 1 Report
Dear authors
I just have a few corrections to ask you.
p1 lane 30 : replace uppercase o with lowercase
p3 lane 128 : there are too many signs
p3 lanes 12ç-130 : I don't understand this sentence can you rephrase it.
Best regards
Author Response
1) P1 lane 30 : replace uppercase o with lowercase
As per the reviewer’s indication, we have replaced uppercase o with lowercase (Page 1, Line 30).
2) P3 lane 128 : there are too many signs
As per the reviewer’s indication, we have deleted the wrong language (Page 3, Line 111-112).
3) P3 lanes 12ç-130 : I don't understand this sentence can you rephrase it.
We are sorry that this sentence made it difficult for the reviewer to understand. We have rephrased this sentence for clarity (Page 3, Line112-114).
Reviewer 2 Report
The manuscript „Current application of iPS cells in the dental tissue regeneration” by Sayuri Hamamo et al. is an interesting review. It is easy to read and follow. It gives you a good overview of the interesting topic of iPS research in the context of dental tissue regeneration. However, in my opinion, several minor changes should be made before this manuscript can be published. See my comments below.
1. The chapters should be rearranged. Chapter two should be transferred to the end of the review. There the authors can discuss the already mentioned problems/limitations and possible solutions, especially in the context of iPS use in dental tissue regeneration.
2. A brief overview of how iPS are generated in general should be included in the text, as well as graphically.
3. The authors mention that the cell source to generate iPS has an influence on their reprogramming and differentiation efficiency. Therefore, in my opinion, it is essential to add the cell source for iPS generation as additional information in each mentioned study throughout the manuscript.
4. MSCs are mentioned several times. Differences to iPS and ESCs should be explained.
5. Third Chapter, line 71: In my opinion, drug discovery is o clinical application, when cells from sick individuals were used to generate iPS. They are a kind of in vitro model, or?
6. Fifth Chapter, line 114-118: Not necessary to explain in detail.
7. Line 128: Wrong language
8. Line 132-133: Periodontal tissues already includes periodontal ligament tissue (PDL).
9. Line 148-149: Sentence not logical. Should be revised.
10. Line 164-171: Not necessary to explain in this detail. These studies have nothing to do with iPS. In my opinion, they can be deleted.
11. Eight chapter: Are there no studies in the context of gingiva regeneration?
12. Line 207: “After the cells were implanted…” Not clear. Which cells?
13. Line 226-230: Authors should provide information on how the cells were differentiated.
14. Line 271-277: Did they also show osteogenesis directly? Otherwise, the conclusion that iPS-MSCs may regenerate alveolar bone during periodontal disease cannot be drawn. It seems that the referenced paper only showed decreased alveolar bone resorption.
15. Several refereneces are missing (e.g. lines 85/86, 88)
Author Response
1) The chapters should be rearranged. Chapter two should be transferred to the end of the review. There the authors can discuss the already mentioned problems/limitations and possible solutions, especially in the context of iPS use in dental tissue regeneration.
Thank you very much for pertinent indication. As per the reviewer’s indication, we have transferred the chapter 2 to the chapter 11. We have also added the relationship with dental tissue regeneration (Page 8, Line 313-337).
2) A brief overview of how iPS are generated in general should be included in the text, as well as graphically.
Thank you very much for pertinent indication. As per the reviewer’s suggestion, we included the text and figure on the generation of iPS cells (Page 1, Line 36-38 and Figure 1).
3) The authors mention that the cell source to generate iPS has an influence on their reprogramming and differentiation efficiency. Therefore, in my opinion, it is essential to add the cell source for iPS generation as additional information in each mentioned study throughout the manuscript.
Thank you very much for appropriate indications. As per the reviewer’s comments, we have included about the cell source of iPS cells (Page 2, Line 72-75). In addition, the source of iPS cells in dental tissue is described in Chapter 3 (Page 2, Line 84-Page 3, Line 94).
4) MSCs are mentioned several times. Differences to iPS and ESCs should be explained.
As per the reviewer’s comment, we have included the differences between MSCs and iPS cells (Page 1, Line 42-43).
5) Third Chapter, line 71: In my opinion, drug discovery is o clinical application, when cells from sick individuals were used to generate iPS. They are a kind of in vitro model, or?
We are so sorry for presenting confusing contents. Drug discovery applying iPS cells is a method to screen drugs using iPS cells in vitro and to develop the drugs efficiently. In other words, iPS cells are indirectly involved in the development of drugs that can be applied clinically. It was difficult to understand in the previous sentence, so we have rephrased this part in revision (Page 2, Line 51-52 and Page 2, Line 61-62).
6) Fifth Chapter, line 114-118: Not necessary to explain in detail.
Thank you very much for precise indication. As per the reviewer’s indication, we have rephrased in short (Page 3, Line 99-100).
7) Line 128: Wrong language
As per the reviewer’s indication, we have deleted the wrong language (Page 3, Line 111-112).
8) Line 132-133: Periodontal tissues already includes periodontal ligament tissue (PDL).
As per the reviewer’s indication, we have corrected this sentence (Page 3, Line 117).
9) Line 148-149: Sentence not logical. Should be revised.
We are sorry that this sentence made it difficult for the reviewer to understand. We have rephrased this sentence for clarity (Page 3, Line132-133).
10) Line 164-171: Not necessary to explain in this detail. These studies have nothing to do with iPS. In my opinion, they can be deleted.
Thank you very much for precise indication. As per the reviewer’s indication, we have rephrased in short (Page 4, Line 147-149).
11) Eight chapter: Are there no studies in the context of gingiva regeneration?
As per the reviewer’s comment, periodontal tissues include not only cementum, alveolar bone, and PDL, but also gingiva. However, there have been no reports of gingival regeneration using iPS cells. Gingiva has a high regenerative capacity. Because there is established treatment to transplant palatal gingiva to the missing part (Zucchelli et al., J Periodontol, 2020), it is thought that gingival regeneration research using iPS cells has not been conducted.
12) Line 207: “After the cells were implanted…” Not clear. Which cells?
We are so sorry for presenting confusing contents. We have clearly described the type of cells that were transplanted (Page 4, Line 183-184).
13) Line 226-230: Authors should provide information on how the cells were differentiated.
Thank you very much for appropriate indications. As per the reviewer’s comments, we have included the information on how the cells were differentiated (Page 5, Line 211; Page 5, Line 214-216; Page 5, Line 220-221; Page 5, Line 228-229, and Page 5, Line 238-239)
14) Line 271-277: Did they also show osteogenesis directly? Otherwise, the conclusion that iPS-MSCs may regenerate alveolar bone during periodontal disease cannot be drawn. It seems that the referenced paper only showed decreased alveolar bone resorption.
Thank you very much for precise indication. As per the reviewer’s indication, this reference suggested that transplantation of iPS-MSCs inhibited alveolar bone resorption. We have deleted this reference because it does not evaluate iPS cells for alveolar bone regeneration (Page 6, Line 248-250).
15) Several references are missing (e.g. lines 85/86, 88)
Because drug discovery research is still at the clinical trial level in Japan, there are no references.